# Relationship between Structure and Biological Activity of Various Vitamin K Forms

**DOI:** 10.3390/foods10123136

**Published:** 2021-12-17

**Authors:** Katarzyna Bus, Arkadiusz Szterk

**Affiliations:** 1Department of Spectrometric Methods, National Medicines Institute, 30/34 Chełmska, 00-725 Warsaw, Poland; 2Center for Translational Medicine, Warsaw University of Life Sciences, Nowoursynowska 100, 02-797 Warsaw, Poland; szterkarkadiusz@gmail.com; 3Transfer of Science Sp. z o.o., Strzygłowska 15, 04-866 Warsaw, Poland

**Keywords:** vitamin K, menaquinone, phylloquinone, bone metabolism, vascular calcification

## Abstract

Vitamin K is involved many biological processes, such as the regulation of blood coagulation, prevention of vascular calcification, bone metabolism and modulation of cell proliferation. Menaquinones (MK) and phylloquinone vary in biological activity, showing different bioavailability, half-life and transport mechanisms. Vitamin K1 and MK-4 remain present in the plasma for 8–24 h, whereas long-chain menaquinones can be detected up to 96 h after administration. Geometric structure is also an important factor that conditions their properties. *Cis*-phylloquinone shows nearly no biological activity. An equivalent study for menaquinone is not available. The effective dose to decrease uncarboxylated osteocalcin was six times lower for MK-7 than for MK-4. Similarly, MK-7 affected blood coagulation system at dose three to four times lower than vitamin K1. Both vitamin K1 and MK-7 inhibited the decline in bone mineral density, however benefits for the occurrence of cardiovascular diseases have been observed only for long-chain menaquinones. There are currently no guidelines for the recommended doses and forms of vitamin K in the prevention of osteoporosis, atherosclerosis and other cardiovascular disorders. Due to the presence of isomers with unknown biological properties in some dietary supplements, quality and safety of that products may be questioned.

## 1. Introduction

Vitamin K is a fat-soluble vitamin characterised by the presence of a 2-methyl-1,4-naphthoquinone ring. There are two naturally occurring vitamin K groups: vitamin K1 (phylloquinone or phytonadione) and vitamin K2 (menaquinone). Vitamin K1 contains a phytyl side chain at the C3 position. It is commonly used as a drug in both injectable and oral formulations for coagulation disorders. Menaquinones are a series of compounds that contain a polyprenyl side chain at the C3 position. They differ in the length of the side chain, which can be denoted as MK-*n*, where *n* is the number of unsaturated β-isoprenoid units in the chain, usually in the range from 4 to 13. Vitamin K also includes a few synthetic forms. Menadione (vitamin K3) has no substituent at the C3 position and acts as a provitamin. It is not recommended for human consumption as it has weak toxicity and may lead to hemolytic anemia and allergic reactions, however it remains in use as a haemorrhage treatment. Menadione salts are used as feed additives for animals [1,2,3]. Vitamin K4 (a group of menadione esters) was found to have cancer fighting properties [4,5]. Vitamin K5 (4-amino-2-methyl-1-naphthol) exhibited antimicrobial and anticancer activity [6,7].

The Vitamin K group (Figure 1) owe their biological activity to the presence of the naphthoquinone ring. It serves as a cofactor for an enzyme that activates vitamin-K-dependent proteins (VKDP), that are involved in processes such as regulation of blood coagulation, prevention of vascular calcification, bone metabolism and modulation of cell proliferation [8,9]. There are differences in the biological activity among various vitamin K forms, despite similar modes of action.

## 2. Biological Functions

### 2.1. Absorption

Vitamin K1 is absorbed mainly in the proximal part of the small intestine, in the presence of bile acids. The process is mediated by the membrane protein Niemann–Pick C1-like 1 (NPC1L1) and the scavenger receptor class B-type I (SR-BI). Both transporters are also involved in cholesterol and vitamin E intestinal uptake. Goncalves et al. (2014) showed that interaction between Vitamin K1 and Vitamin E at the level of intestinal absorption may induce excessive bleeding observed in patients supplemented with vitamin E [10]. In another study, Takada et al. (2015) found that the coadministration of ezetimibe (NPC1L1-selective inhibitor used for dyslipidemia) and warfarin (Vitamin K antagonist) reduced hepatic vitamin K level and enhanced the anticoagulant effect of warfarin [11].

There are no data on the molecular mechanism of intestinal vitamin K2 absorption. It has been indicated however that menaquinones are absorbed by a passive diffusion in the ileum and colon and that the process is modified by bile salt concentration, the presence of unsaturated fatty acids and the luminal pH [12].

### 2.2. The Vitamin K Cycle

The active form of vitamin K is hydroquinone. It is produced from quinone under the influence of cytoplasmic quinone reductase (QR1) or vitamin K epoxide reductase (VKOR), an enzyme located in the endoplasmic reticulum membrane [13,14,15]. Hydroquinone acts as a cofactor for γ-glutamylcarboxylase (GGCX), and in this reaction vitamin K is oxidised to 2,3-epoxide. The last stage is restoration of the quinone molecule (Figure 2). These transformations, called the vitamin K cycle, enable multiple uses of a single molecule and determine the activation of γ-carboxy-glutamic acid (Gla) protein even with low intake of vitamin K [16,17,18]. VKOR plays a factor in the prevention of vitamin K deficiency, as demonstrated in an in vivo study in rats. Mutation of this enzyme at the vitamin K binding site led to artery calcification in rats that had received a vitamin K-deficient diet, while in rats with normal VKOR there were no vessel changes when given the same diet. Additionally, the diet supplemented with vitamin K prevented vessel calcification despite the VKOR mutation [19].

### 2.3. The Role of GGCX

GGCX plays an essential role in post-translational VKDP modification by converting glutamic acid (Glu) residues to Gla. Gla in proteins increases their affinity for calcium ions, which in turn determines their biological activity. GGCX is a dual-function membrane protein located in the rough endoplasmic reticulum. It consists of five transmembrane domains. GGCX carboxylates the Glu substrate, by removing γ-hydrogen and adding CO_2_, while it simultaneously oxidises the vitamin K hydroquinone to 2,3-epoxide [20]. GGCX gene mutations are mainly associated with deficiency of blood coagulation factors and often have fatal outcomes. However, in patients with less severe blood coagulation disorders, there is an additional phenotype not related to bleeding. Its occurrence includes symptoms such as premature osteoporosis, midfacial hypoplasia and mineralisation of skin that leads to pseudoxanthoma-elasticum-like disorder. These diseases stem from deficiency of osteocalcin (OC) and matrix Gla protein (MGP) [21,22,23].

### 2.4. VKDPs

Seventeen GGCX-activated proteins have been identified. There are three VKDP groups. One group includes proteins such as OC, MGP and growth arrest-specific 6 protein (Gas6) which are produced extrahepatically and are not related to the blood coagulation process [9,24,25]. Subsequent Gla-proteins: periostin and periostin-like factor are expressed under the influence of an injury and participate in tissue remodelling [26,27,28]. There are also transmembrane and proline-rich Gla proteins, but their functions are unknown [29,30]. The second group consists of proteins produced in the liver that are involved in blood coagulation. These include blood coagulation factors II (prothrombin), VII, IX and X. The last group includes anticoagulants, also produced in the liver: C, S and Z proteins.

In the normal adult population, approximately 20–30% of OC and MGP are present in a non-carboxylated form. Gla proteins responsible for the blood coagulation process, in contrast to extrahepatic Gla proteins, are fully carboxylated [31]. Therefore, both vitamin K1 and K2 supplementation under physiological conditions does not affect blood coagulation factors but it increases the level of carboxylated extrahepatic VKDP [32,33,34,35,36]. However, in the case of vitamin K deficiency, e.g., in haemodialysis patients, menaquinone-7 (MK-7) supplementation reduced the level of both uncarboxylated extrahepatic OC and MGP (ucOC and ucMGP) and those produced in the liver—prothrombin (PIVKA-II, protein induced by vitamin K absence II) [37]. 

### 2.5. Effects on Bone Metabolism

An inactive form of OC (uncarboxylated OC or ucOC) is synthesised by osteoblasts during bone formation. With the participation of reduced vitamin K, ucOC is modified to a carboxylated form (Gla-OC or cOC) that has higher affinity for calcium ions. Gla-OC is secreted into the bone matrix, where it is one of the main non-collagenous bone proteins [38]. OC, along with osteopontin, is involved in dilatational band formation, which affects bone mechanical properties [39,40]. Additionally, in vitro studies on mesenchymal stromal cells (MSCs) and in vivo studies with OC-deficient mice showed that reduced OC level delayed mineral maturation [41,42]. Besides a direct effect on bone matrix, undercarboxylated OC acts also as a hormone on beta cells, adipocytes and Leydig cells [43,44]. OC is a bone remodelling marker; total serum OC or the cOC/ucOC ratio is quantified for the diagnosis of osteoporosis [45].

Additionally, vitamin K2 regulates bone metabolism through mechanisms not associated with OC activation. MK-7 induces expression of alkaline phosphatase (ALP), Runx2 and Osterix in osteoblast precursor cells and also stimulates their differentiation into mineralising osteoblasts. The opposite effect occurs in osteoclast precursor cells: vitamin K2 reduces cell differentiation and bone resorption. The mechanism of this action is related to inhibition of cytokine-induced nuclear factor-kappa B (NF-κB) activation. Indeed, MK-7, and to a lesser extent MK-4, inhibited NF-κB, while phylloquinone displayed no such activity [46,47]. Moreover, vitamin K2, through the activation of pregnane X receptor (PXR or steroid and xenobiotic receptor SXR), regulates transcription of genes that encode extracellular matrix (ECM) proteins and therefore promotes collagen synthesis in osteoblasts [48,49]. The effect of vitamin K2 on bone metabolism is shown in Figure 3.

### 2.6. Effects on Vascular Calcification

MGP is produced in vascular smooth muscle cells (VSMCs) and chondrocytes. MGP binds calcium ions, which prevents their deposition in soft tissues and inhibits calcification [24,50]. Protein activation requires the presence of vitamin K and then, in order to increase secretion, MGP undergoes additional post-translational modification based on serine phosphorylation [51]. In blood vessel walls, active MGP prevents VSMC transdifferentiation to osteoblasts by inhibiting the activity of bone morphogenetic proteins 2 and 4 (BMP-2, BMP-4) [50,52]. Abnormal level of active MGP, that results from vitamin K deficiency, is associated with vessel and valve calcification. The use of VKOR inhibiting anticoagulant correlates with increased risk of soft tissue calcification. The study had shown that this process was mediated by MGP [53]. MGP affects atherosclerotic plaques. This phenomenon occurs directly by binding calcium ions and indirectly by suppressing BMP activity. The overall results are the inhibition of calcification and inflammation in affected areas [50,52]. The mRNA expression of MGP is upregulated by vitamin D3 at physiological concentrations. Therefore, the low vitamin D3 levels may be responsible for the increased risk of vascular calcification e.g., in patients with chronic kidney disease [54]. MGP expression is also enhanced by oestrogen through inhibition of the vascular RANKL signalling pathway [55].

### 2.7. Effects on Cell Proliferation

Gas6 affects adipocyte, VSMC, endothelial and bone marrow cell functions by regulating proliferation, cell differentiation and the release of inflammatory mediators. Gas6 activity is mediated by TAM receptors (Axl, Tyro3 and MerTK), which belong to the protein-tyrosine kinase family. Impaired Gas6 function may serve as an risk factor for atherosclerosis, cancer as well as metabolic and autoimmune diseases [56,57]. It has been shown that vitamin K2, by activating Gas6, exerts a protective effect on VSMCs and prevents their apoptosis [24].

### 2.8. Effects on Neural Cells

Vitamin K exhibits neuroprotective activity through mechanisms unrelated to GGCX activity. Sakaue et al. (2011) have shown that both vitamin K1 and MK-4 inhibited methylmercury-induced neuronal death. The protective mechanism included an effect against the degradation of intracellular glutathione (GSH), which is related to the pathogenesis of neurodegenerative disorders [58]. A recent study has shown that vitamin K2 decreased the beta-amyloid and H_2_O_2_ cytotoxicity in neuroblastic cells [59]. This is consistent with previous results and indicates the beneficial role of vitamin K2 against the progression of Alzheimer’s disease.

MK-4 was also found to act as a mitochondrial electron carrier. It rescued mitochondrial dysfunction related to PINK1 mutation observed in Parkinson’s disease [60]. In microglial cell line study, MK-4 was found to repress reactive oxygen species production and NF-κB signalling pathway activation [61]. These results reveal a role of MK-4 in the reduction of oxidative stress and neuroinflammation, which are involved in neuronal degeneration in Parkinson’s disease.

## 3. Daily Intake and Dietary Sources

Due to vitamin K’s significance in the human body, daily diet should include an adequate amount of both vitamin K1 and K2. There are no specific guidelines for daily menaquinone amount requirements. Current vitamin K recommendations were established based on phylloquinone and its antihemorrhagic effects. For infants and children, the daily intake should be 1 μg/kg body weight, and the recommended daily dose for an adult is 70 μg. It is assumed that the average daily intake of vitamin K1 and K2 in total is 70–200 µg, for various European countries. Based on the consumption of vitamin-K-rich food, it is estimated that the daily supply of menaquinones is 10–35 μg, which is an average of 25% of the total amount of vitamin K in the diet [8,62,63,64,65,66]. This amount may be insufficient to ensure proper bone mineralisation or VSMC function. Knapen et al. (2013) and Forli et al. (2010) proved that increased vitamin K2 intake prevented osteoporosis by increasing bone mineralisation. In these studies, 180 µg MK-7 supplements were administered for 3 years to a group of healthy postmenopausal women and patients after lung or heart transplants. The study findings confirmed the beneficial effect of a MK-7 dose higher than the daily supply in the prevention of diseases such as osteoporosis, especially in high-risk groups [67,68]. 

The main vitamin K sources in the human diet are plants and dairy products. Green vegetables, such as spinach, broccoli and cabbage, contain mainly phylloquinone. Other sources include vegetable oils: soybean, rapeseed and olive oil and products derived from them, such as margarine. Menaquinones are supplied mainly through milk and poultry products. They contain vitamin K2 in the form of MK-4, which is biosynthesized from phylloquinone added to animal feed. Menaquinones with longer isoprenoid chains, such as MK-6, MK-7 and MK-9, are produced by bacteria. Therefore, the high content of long-chain menaquinones is characteristic for fermented products such as cheese, curd and sauerkraut. An important source of MK-7 is natto, made from soybeans fermented with *Bacillus subtilis* and it is a traditional ingredient in Japanese cuisine. Menaquinones are also produced by human intestinal microflora, but their bioavailability is low [69,70,71,72]. 

Methods used for industrial MK-7 production include biological (soybean fermentation with *B*. *subtilis*) and chemical synthesis. MK-7 is the main product synthesised by *B*. *subtilis*, and it accounts for over 95% of the total amount of all formed menaquinones [72,73]. Chemical synthesis is a more economical method. Obtaining high-yield all-*trans* MK-7 and purifying it from by-products such as *cis* isomers and menaquinones with different side chain length is problematic [74,75,76]. Purification of the *trans* isomer is crucial because of the possible differences in biological activity of various isomers. In the case of phylloquinone, the *cis* form had 1% of the biological activity relative to the *trans* form [77,78,79]. 

## 4. Differences between Phylloquinone and Menaquinone Biological Activity

### 4.1. Absorption and Tissue Distribution

Vitamins K1 and K2 have similar biological functions, but they differ in tissue distribution. High phylloquinone levels occur in the liver, heart and pancreas, but lower concentrations are also present in the brain, kidneys and lungs. MK-4 is found in all tissues, but the concentration in the liver, heart and lungs is low. MK-4 levels are higher or similar to K1 in the brain, kidneys and pancreas. Long-chain menaquinones are present in the liver and their total level is higher than K1 [80]. 

Subsequent differences between the K group vitamins are apparent in bioavailability studies. Vitamin K1 and MK-4 were present in the plasma for 8–24 h after administration. The half-life of long-chain menaquinones (MK-7 and MK-9) was much longer; they were detected up to 96 h after administration [81,82]. Additionally, during long-term supplementation, MK-7 accumulated in the serum and reached a concentration several times higher than K1 [81]. Bioavailability is also dependent on vitamin K source. The amount of absorbed vitamin K1 from vegetable products was much lower than the equivalent dose given as a supplement [83]. On the other hand, MK-7 from natto demonstrated greater bioavailability compared with K1 from both cooked spinach and a K1 supplement [84].

Differences between bioavailability and half-life may arise from various vitamin K1 and menaquinone transport mechanisms. Vitamin K1 was mainly transported in lipoproteins rich in triacylglycerols (TGRLP), and to a lesser extent in high-density lipoproteins (HDL) and low-density lipoproteins (LDL). MK-4 was present in all fractions in equal concentrations. MK-9 was initially present in the triacylglycerol fraction, and 8 h after administration it was also present in LDL [85]. These results explain the long blood half-life of long-chain menaquinones, which are initially transported to the liver with TGRLP, from where they are released in the LDL to the target tissues.

### 4.2. Relationship between Activity and Chain Length

An in vitro study of vitamin K cofactor activity on hepatic carboxylase was performed. The results obtained for menaquinones that contain two to seven isoprene units were comparable to phylloquinone. As side chain length increased, the required vitamin K concentration to achieve half-maximal reaction velocity decreased. However, the reaction rate at saturating vitamin concentrations also declined. Menadione and MK-1 showed no effect, or the activity was very low [86]. The relationship between activity and isoprene chain length was also investigated in in vivo studies. Serum ucOC concentration significantly decreased after administration of 100 μg MK-7 [87], while in the case of MK-4, the effective dose was 600 μg [88]. Vitamin K1 and MK-7 administered at 0.22 µmol/day (approximately 100 and 140 μg/day, respectively), initially both increased the cOC/ucOC ratio, but only MK-7 maintained that effect during the entire experiment [81].

### 4.3. Relationship between Activity and Double Bond Configuration

As previously mentioned, the biological activity also depends on the configuration of double bonds in the side chain. Studies in rats revealed that the cis vitamin K1 isomer had practically no biological activity, but the tissue distribution of cis-phylloquinone was similar to the distribution of the trans form. However, the exception was a liver, where the amount of cis isomer was greater than the trans [79]. Another study showed differences in metabolism: the concentration of the cis isomer of vitamin K1 in liver decreased much slower than the trans one. Moreover, the cis isomer was associated with the mitochondrial fraction and the trans isomer with the endoplasmic reticulum fraction, where the vitamin K cycle takes place [77]. To date, there are no studies on the biological activity of the geometric menaquinone isomers. However, based on the similarity in action and structure to phylloquinone, it can be assumed that their activity is dependent on the presence of trans bonds.

### 4.4. Effect on Blood Coagulation

Vitamin K1 is commonly used to prevent haemorrhagic disease in new-borns [66]. Studies have shown that menaquinones also affect proper functioning of the blood coagulation system. In Japan, MK-4 was successfully used to prevent vitamin K deficiency in new-borns [89]. Schurgers et al. (2007) showed that MK-7, at dose three to four times lower than vitamin K1, reduced INR in healthy volunteers who received the vitamin K agonist acenocoumarol [81]. These results are consistent with the previously described in vitro and in vivo studies on GGCX activity. In healthy volunteers however MK-7 supplementation did not affect blood coagulation [32,33]. This outcome may be due to the fact that the coagulation factors, under physiological conditions, are completely carboxylated, and the increased supplementation of vitamin K does not interfere with the blood coagulation process. In subsequent studies, in contrast to previous data, MK-4 supplementation increased PT-INR within normal limits [88]. The results are not unequivocal, but it is important to note that supplementing with phylloquinone as well as menaquinones does affect anticoagulant therapy and this fact should be taken into account before starting supplementation.

### 4.5. Osteoporosis Prevention

Studies performed in recent years demonstrated that vitamin K has a positive effect on bone mineralisation and the prevention of age-related osteoporotic lesions. The highest risk of osteoporosis occurs in postmenopausal women; therefore, they make up a large group of study participants. Knapen et al. (2007) investigated the supplementation of 45 mg MK-4 daily in comparison to placebo over 3 years. Bone mineral content (BMC) and femoral neck width (FNW) increased in the MK-4 group; however, despite the high dose, bone mineral density (BMD) remained unchanged [90]. In another study, MK-7 was administered at 180 μg/day. Compared to the control group, 3 years of supplementation inhibited the decline in BMC and BMD at the lumbar spine and femoral neck [68] Kanellakis et al. (2012) examined the effect of a 12-month supplementation with 100 μg/day vitamin K1 or 100 μg/day MK-7, both in combination with calcium and vitamin D3. Lumbar spine BMD increased in the groups that received vitamin K1 or MK-7, but this parameter remained unchanged in the control group that received only calcium and vitamin D3 [91]. Nonetheless, vitamin K2 supplementation is a preventative measure rather than an osteoporosis treatment. Meta-analysis of various interventions for improving BMD revealed, that vitamin K2 can increase lumbar spine BMD. It ranked sixth among the eighteen different single or combined interventions including Ca, vitamin D, estrogen, isoflavone and exercise [92].

### 4.6. Cardiovascular Disease Prevention

A few different studies investigated the dependence of dietary phylloquinone and menaquinones intake on the occurrence of cardiovascular diseases. Beneficial effects have been observed only for long-chain menaquinones. In population research, Geleijnse et al. (2004) showed a connection between higher estimated menaquinone intake (above 21.6 μg/day) and decreased coronary heart disease-related mortality and aortic calcification, but there was no such correlation for phylloquinone [93]. In subsequent study, Beulens et al. (2009) showed that the dietary intake of menaquinones (MK-4 through MK-10, 48.5 μg/day), but not phylloquinone, was associated with a reduced risk of coronary calcification. The study was conducted on a group of postmenopausal women, and the amount of dietary vitamin K was evaluated with a questionnaire [94].

## 5. Conclusions

Vitamin K is implicated in a number of essential biological processes. The lipophilic properties of phylloquinone and menaquinones result in their varying biological activity. The compounds exhibit different bioavailability, half-life and transport mechanisms. Geometric structure is also an important factor that conditions their properties. Cis-phylloquinone exhibits very little biological activity, but there are no such studies for menaquinones. 

Numerous clinical trials have shown menaquinone’s preventive action in osteoporosis and cardiovascular diseases, but usually in doses higher than the estimated daily intake. These results led to the emergence of many dietary supplements containing MK-7, the most-studied form of menaquinone. However, the findings are still insufficient to establish guidelines for the recommended doses and forms of vitamin K to be supplemented in the prevention of osteoporosis, atherosclerosis and other cardiovascular disorders. Because of the presence of isomers with unknown biological properties in some products, their quality, safety and effectiveness may be questionable.

It is well established that vitamin K interacts with anticoagulants such as warfarin, limiting preventative supplementation in patients with thromboembolism. According to other scientific reports, the Gas6/Axl pathway plays a key role in mesangial cell proliferation. Inhibition of that pathway, e.g., by warfarin, may be a potentially important therapeutic target for the treatment of glomerulonephritis [95]. However, no data are available to prove whether vitamin K supplementation may induce kidney failure. The use of vitamin K in patients with comorbidities should be cautionary.

## Figures and Tables

**Figure 1 foods-10-03136-f001:**
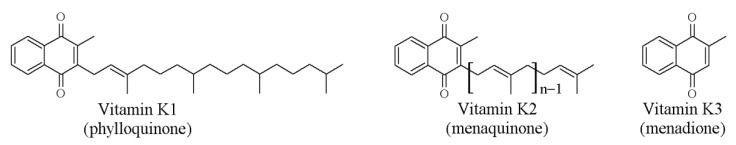
Chemical structures of different vitamin K forms.

**Figure 2 foods-10-03136-f002:**
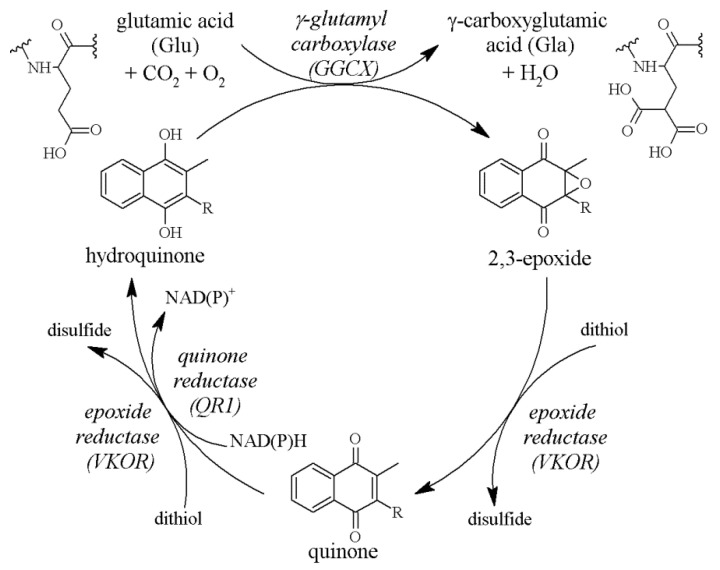
The vitamin K cycle. Vitamin K (quinone) is reduced to hydroquinone by QR1 or VKOR. Then, during carboxylation of Glu to Gla, it serves as an electron donor and undergoes oxidation to 2,3-epoxide. Finally, it is converted back to quinone by VKOR. Abbreviations: QR1, quinone reductase; VKOR, vitamin K epoxide reductase; Glu, glutamic acid; Gla, γ-carboxyglutamic acid.

**Figure 3 foods-10-03136-f003:**
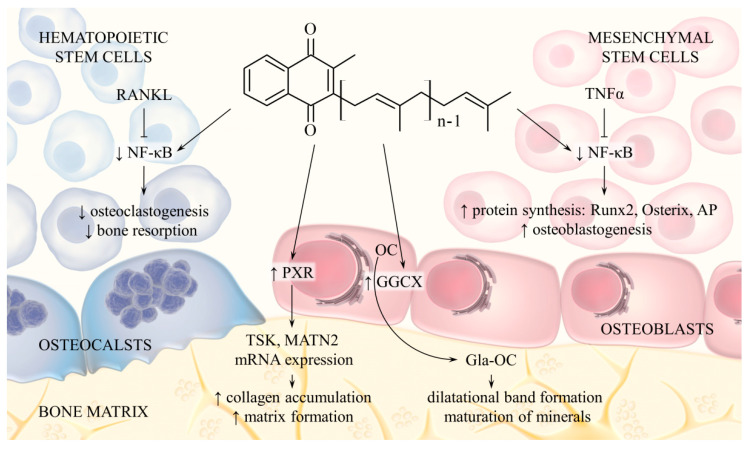
Menaquinone pathways of action associated with bone metabolism. Menaquinone is cofactor for GGCX, which catalyses activation of OC. Vitamin K2 also inhibits NF-κB activation induced by cytokines (RANKL, TNFα), which leads to inhibition of osteoclastogenesis and activation of osteoblastogenesis. Another mechanism includes activation of nuclear receptor PXR in osteoblasts, which increases expression of genes encoding bone matrix proteins. Abbreviations: RANKL, receptor activator of nuclear factor kappa-Β ligand; TNFα, tumour necrosis factor alpha; NF-κB, nuclear factor-kappa B; AP, alkaline phosphatase; PXR, pregnane X receptor; OC, osteocalcin; Gla-OC, carboxylated osteocalcin.

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
