# Peer review of "Relationship between Structure and Biological Activity of Various Vitamin K Forms"

_foods, 2021, doi:10.3390/foods10123136_

Round 1

Reviewer 1 Report

This manuscript has been well summarized about the relationship between chemical structure and difference of bioactivity of vitamin K. But authors need to revise the manuscript in following points. 

Authors have to touch intestinal absorption system of vitamin K via NPC1L1. This transporter can transport not only vitamin K1 but also cholesterol, tocopherol.

Line 178 on page 5
"Natto is a traditional ingredient in Japanese cuisine and its common in the Asian diet"
Natto is only eaten in Japan, not in other Asian counties so much.
Authors should remove the word "and its common in the Asian diet" from the manuscript.

Line 181 on page 5
"industrial vitamin K2 production" should be changed to "industrial MK-7 production".

Reviewer 2 Report

Ambitious report, some comments in enclosed PDF.

Reviewer 3 Report

STREGHT: in this paper some review about vitamin k was studied and interesting graph are proposed. Daily intake absorption rates  of Menaquinones were studied plus  Cis-phylloquinone. The demonsstrated vitamin K2 preventive activity on diseases the most-studied menaquinone. some guidelines for the recommended doses and forms of vitamin K that should be supplemented in the prevention of osteoporosis, atherosclerosis and other cardiovascular disorders are commented .

Weakness please correct abstract put some results there

2 please check accurately tipos in english
